# Smoothness Matrices Beat Smoothness Constants: Better Communication Compression Techniques for Distributed Optimization

**Mher Safaryan**
KAUST, Saudi Arabia
mher.safaryan.1@kaust.edu.sa

**Filip Hanzely**
TTIC, United States
fhanzely@gmail.com

**Peter Richtárik**
KAUST, Saudi Arabia
peter.richtarik@kaust.edu.sa

## Abstract

Large scale distributed optimization has become the default tool for the training of supervised machine learning models with a large number of parameters and training data. Recent advancements in the field provide several mechanisms for speeding up the training, including *compressed communication*, *variance reduction* and *acceleration*. However, none of these methods is capable of exploiting the inherently rich data-dependent smoothness structure of the local losses beyond standard smoothness constants. In this paper, we argue that when training supervised models, *smoothness matrices*—information-rich generalizations of the ubiquitous smoothness constants—can and should be exploited for further dramatic gains, both in theory and practice. In order to further alleviate the communication burden inherent in distributed optimization, we propose a novel communication sparsification strategy that can take full advantage of the smoothness matrices associated with local losses. To showcase the power of this tool, we describe how our sparsification technique can be adapted to three distributed optimization algorithms—DCGD [Khirirat et al., 2018], DIANA [Mishchenko et al., 2019] and ADIANA [Li et al., 2020]—yielding significant savings in terms of communication complexity. The new methods always outperform the baselines, often dramatically so.

## 1 Introduction

With the desire to build and train high quality machine learning models comes an increased appetite for larger models, both in terms of the number of parameters encoding them, and in the amount of data required to train them. In the big data regime, the data is partitioned among many parallel machines, which then cooperatively train a single global model, usually orchestrated by a central server. Distributed training is cast as the distributed optimization problem

$$\min_{x \in \mathbb{R}^d} f(x) + R(x), \qquad f(x) \coloneqq \frac{1}{n} \sum_{i=1}^{n} f_i(x), \tag{1}$$

where $d$ is the number of parameters of model $x \in \mathbb{R}^d$, $n$ is the number of machines participating in the training, $f_i(x)$ is the loss associated with the data stored on machine $i \in [n] \coloneqq \{1, 2, \ldots, n\}$, $f(x)$ is the empirical loss, and $R(x)$ is a regularizer. Ample research over the past two decades has shown that first-order methods are highly scalable and as a result are the methods of choice for distributed optimization problems [Liu and Zhang, 2020]. In particular, a substantial amount of work

35th Conference on Neural Information Processing Systems (NeurIPS 2021).

has been devoted to speeding up the training process by developing efficient methods empowered with techniques such as *compressed communication*, *variance reduction* and *acceleration*.

**Compressed communication.** In distributed training, compute nodes have to communicate with each other, often via a central server, in order to be able to maintain consensus and jointly train a global model. However, communication of the information pertaining to local progress, which is typically contained in gradient(s) distilled from local data, is almost invariably *the* key bottleneck in distributed training systems [Xu et al., 2020]. One popular way to address this issue is to reduce the number of bits encoding the vector/tensor to be transferred via the help of a lossy *compression operator*. Numerous *unbiased* gradient compression operators have been proposed for this purpose, including several types of sparsifications [Wang et al., 2018, Mishchenko et al., 2020, Alistarh et al., 2018] and quantizations [Alistarh et al., 2017, Zhang et al., 2017, Horváth et al., 2019a, Wu et al., 2018].

**Variance reduction.** A marked issue that needs to be addressed by successful distributed optimization methods has to do with the (potential) "dissimilarity" of the local loss functions $f_1, \ldots, f_n$, which in turn is due to the heterogeneity of the training data defining these functions. The higher the dissimilarity, the harder it is for the devices to find the minimizer of (1). This issue exists even in the unregularized case ($R \equiv 0$). Indeed, while in this case $\frac{1}{n} \sum_i \nabla f_i(x^*) = 0$ if $x^*$ is a minimizer of $f$, this does not mean that the individual gradients, $\nabla f_1(x^*), \ldots, \nabla f_n(x^*)$, are all zero. This shows that local gradient information alone is not enough for any node to "realize" that a solution has been found, which encourages further, in this case unnecessary, iterations. If unaddressed properly, an algorithm is forced to use smaller learning rates, and this leads to unnecessarily slow convergence. On the other hand, when a fixed learning rate is used, the rate is fast, but convergence stops in a potentially large neighborhood[1] of the optimum $x^*$. This issue is exacerbated further by the extra noise coming from gradient compression. Indeed, this noise prevents methods such as Distributed Compressed Gradient Descent (DCGD) [Khirirat et al., 2018] from converging to $x^*$ with a constant learning rate even in the interpolation regime characterized by the identities $\nabla f_i(x^*) = 0$ for all $i$. Fortunately, these issues can be resolved via carefully designed variance reduction techniques [Gower et al., 2020]. In particular, the first variance reduction mechanism for removing the variance coming from compression operators in distributed training is due to Mishchenko et al. [2019], embodied in their DIANA algorithm. The method was initially analyzed for ternary quantization only [Wen et al., 2017], and later generalized to handle a general class of unbiased compression operators [Horváth et al., 2019b, Gorbunov et al., 2020b].

**Acceleration.** To speed up distributed training even further, it is often possible to employ Nesterov's acceleration technique [Nesterov, 1983, 2004] in concert with gradient compression and variance reduction. For instance, Li et al. [2020] developed the ADIANA method, which adds acceleration on top of a variant of DIANA that relies on the computation of full-batch gradients on all nodes. The resulting method offers provable speedups in convex and strongly convex regimes. Another example is the method ECLK of Qian et al. [2020], which employs compressed communication via any (possibly biased) compressor satisfying a certain contraction property in combination with a slightly different variance reduction technique known as error compensation [Stich and Karimireddy, 2019, Karimireddy et al., 2019], while acceleration is offered by a loopless variant of the accelerated method Katyusha [Allen-Zhu, 2017, Kovalev et al., 2020].

**Further tricks.** Numerous other techniques are often used to improve some other aspects of distributed training, including implementing multiple local gradient steps before communication [Stich, 2020, Karimireddy et al., 2020, Woodworth et al., 2020a], asynchronous communication protocols [Agarwal and Duchi, 2011, Lian et al., 2015, Recht et al., 2011], in-network aggregation [Sapio et al., 2021], and performing the distributed training in a decentralized peer-to-peer manner without the reliance on an orchestrating server [Koloskova et al., 2019, Alghunaim et al., 2019]. However, in this work, we do not explore these directions and focus on the three techniques described before, namely, compressed communication, variance reduction and acceleration.

## 2 Mining for Smoothness Information

**2.1. One size fits all.** Arguably, one of the most ubiquitous, if not *the* most ubiquitous, assumptions used in the literature on first-order optimization methods is that of *L-smoothness* [Nesterov, 2004]. A

---

[1]In the $R \equiv 0$ case, this neighborhood is proportional to the *variance of the local gradients at the optimum*: $\frac{1}{n} \sum_{i=1}^{n} \|\nabla f_i(x^*)\|^2$.

differentiable function $\phi: \mathbb{R}^d \to \mathbb{R}$ is said to be $L$-smooth if there exists a constant $L \geq 0$ such that

$$\phi(x) \leq \phi(y) + \langle \nabla \phi(x), x - y \rangle + \frac{L}{2} \|x - y\|^2 \tag{2}$$

holds for all $x, y \in \mathbb{R}^d$. However, most works in the area of finite-sum distributed optimization use it very crudely: they assume that all local loss functions $f_i$ as well as their average, $f = \frac{1}{n} \sum_i f_i$, share the same smoothness constant $L$ [Tang et al., 2019, Woodworth et al., 2020b, Stich, 2020]. This is crude because much information is lost. Indeed, assuming that each $f_i$ is $L_i$-smooth, it is well known that $f$ is $L_f$-smooth with $L_f$ satisfying $L_f \leq \frac{1}{n} \sum_i L_i$. In the light of this, the above assumption is crude as it effectively replaces the values $L_1, \ldots, L_n$ and $L_f$ with a single parameter $L$ satisfying $L \geq \max\{L_1, \ldots, L_n\}$. Since the stepsizes and convergence rates of first-order methods depend on the smoothness constant(s) employed, convergence analysis relying on such crude approximation may be significantly suboptimal, and the methods too slow when implemented following the theory.

**2.2. "According to the work of their hands" (Lam 3:64).** Significant theoretical and practical improvement can often be obtained when taking account of all the smoothness constants involved, avoiding the practice of replacing them all with a single crude bound. Such analyses are more rare, but fairly common. For example, [Richtárik and Takáč, 2016a, Hanzely and Richtárik, 2019a].

**2.3. "Like treasure hidden in a field, which a man found and covered up" (Mat 13:44).** The starting point of this paper is the observation that there is a hitherto untapped richness of smoothness information that *can* be used to construct *better distributed optimization algorithms and obtain better theory.* This information is available, but hidden from sight, and is based on the notion of *matrix smoothness*.

**Definition 1** (Matrix Smoothness). We say that a differentiable function $\phi: \mathbb{R}^d \to \mathbb{R}$ is $\mathbf{L}$-smooth if there exists a symmetric positive semidefinite matrix $\mathbf{L} \succeq 0$ such that

$$\phi(x) \leq \phi(y) + \langle \nabla \phi(y), x - y \rangle + \frac{1}{2} \|x - y\|_{\mathbf{L}}^2, \quad \forall x, y \in \mathbb{R}^d. \tag{3}$$

The standard $L$-smoothness condition (2) is obtained as a special case of (3) for matrices of the form $\mathbf{L} = L\mathbf{I}$, where $\mathbf{I}$ is the identity matrix. In particular, if function $f_i$ appearing in (1) is often the average loss over the training data stored on node $i$, i.e.,

$$f_i(x) = \frac{1}{m_i} \sum_{m=1}^{m_i} \phi_{im}(\mathbf{A}_{im} x), \tag{4}$$

where $\mathbf{A}_{im} \in \mathbb{R}^{d_{im} \times d}$ is a data matrix, and $\phi_{im}: \mathbb{R}^{d_{im}} \to \mathbb{R}$ is a differentiable function (e.g., the loss over all but the last linear layer of a NN). The following simple result from Qu and Richtárik [2016b], used therein in the context of randomized coordinate descent methods, states that if the loss functions $\phi_{im}$ are smooth in the standard scalar sense, then $f_i$ is smooth in the matrix sense.

**Lemma 1.** *If each $\phi_{im}$ is $\lambda_{im}$-smooth, then the function $f_i$ defined in* (4) *is $\mathbf{L}_i$-smooth with*

$$\mathbf{L}_i = \frac{1}{m_i} \sum_{m=1}^{m_i} \lambda_{im} \mathbf{A}_{im}^\top \mathbf{A}_{im}. \tag{5}$$

In cases where the local functions $f_i$ are of the form (4)[2]—and it is clear this structure is ubiquitous— there is a lot of potentially useful information contained in the matrix smoothness "constant" $\mathbf{L}_i$. If we were to use the scalar smoothness constant of $f_i$ instead, we would be effectively tossing this richness away, and replacing it with $L_i = \lambda_{\max}(\mathbf{L}_i)$; the largest eigenvalue of $\mathbf{L}_i$. This seems wasteful. As we show in this work, it is. However, we offer a fix.

## 3 Motivation and Contributions

To the best of our knowledge, *none* of the current distributed optimization methods, including the methods DCGD [Khirirat et al., 2018], DIANA [Mishchenko et al., 2019] and ADIANA [Li et al., 2020] discussed in Section 1, are capable of exploiting the inherently rich data-dependent smoothness structure of the local losses beyond standard smoothness constants. To this effect, we impose the following assumption throughout the paper:

**Assumption 1.** The functions $f_i: \mathbb{R}^d \to \mathbb{R}$ are differentiable, convex, lower bounded[3] and $\mathbf{L}_i$-smooth. Moreover, $f$ is $\mathbf{L}$-smooth with (standard) smoothness constant $L := \lambda_{\max}(\mathbf{L})$.

---

[2] Our theoretical results hold for general loss functions $f_i$ and do not assume the structure (4)

[3] Lower boundedness of $f_i(x)$ can be dropped if $\mathbf{L}_i \succ 0$ is positive definite. This part of the assumption is not a restriction in applications as all loss function are lower bounded.

**Table 1:** Original and proposed new methods.

| ORIGINAL | DCGD | DIANA | ADIANA |
|---|---|---|---|
| **NEW** | DCGD+ (ALG.1) | DIANA+ (ALG.2) | ADIANA+ (ALG.3) |
| PROXIMAL | ✓ | ✓ | ✓ |
| DISTRIBUTED | ✓ | ✓ | ✓ |
| VARIANCE REDUCED | ✗ | ✓ | ✓ |
| ACCELERATED | ✗ | ✗ | ✓ |

**Table 2:** Summary of theoretical results obtained in this work with hidden $\log \frac{1}{\varepsilon}$ factors and constants. Below $n$ is the number of machines, $d$ is the number of parameters of model, $L_{\max} = \max_i L_i$, $L_i = \lambda_{\max}(\mathbf{L}_i)$ and the expected smoothness constant $\widetilde{\mathcal{L}}_{\max}$ is defined in (9). The variance of generic compression operator used in the original methods is denoted by $\omega$. In case of sparsification, we have $\omega = d/\tau - 1 = \mathcal{O}(n)$ when the expected size of selected coordinates is $\tau = d/n$. Parameters $\nu_1, \nu_2$ and $\nu$ describing matrices $\mathbf{L}_i$ are defined in (13). See Table 6 for further notations.

| Regime | $\nabla f_i(x^*) \equiv 0$ | arbitrary $\nabla f_i(x^*)$ | arbitrary $\nabla f_i(x^*)$ |
|---|---|---|---|
| **Original Methods** | **DCGD** [Khirirat et al., 2018] | **DIANA** [Mishchenko et al., 2019] | **ADIANA** [Li et al., 2020] |
| Iteration Complexity | $\frac{L}{\mu} + \frac{\omega L_{\max}}{n\mu}$ | $\omega + \frac{L_{\max}}{\mu} + \frac{\omega L_{\max}}{n\mu}$ | $\begin{cases} \omega + \omega\sqrt{\frac{L_{\max}}{n\mu}} & \text{if } n \leq \omega \\ \omega + \sqrt{\frac{L_{\max}}{\mu}} + \sqrt{\omega\sqrt{\frac{\omega L_{\max}}{n\mu}}\sqrt{\frac{L_{\max}}{\mu}}} & \text{if } n > \omega \end{cases}$ |
| Iteration Complexity $\tau = d/n$ | $\frac{L_{\max}}{\mu}$ | $n + \frac{L_{\max}}{\mu}$ | $n + n\sqrt{\frac{L_{\max}}{n\mu}} \equiv n + \sqrt{n\frac{L_{\max}}{\mu}}$ |
| **New Methods** | **DCGD+** (Algorithm 1) | **DIANA+** (Algorithm 2) | **ADIANA+** (Algorithm 3) |
| Iteration Complexity | $\frac{L}{\mu} + \frac{\widetilde{\mathcal{L}}_{\max}}{n\mu}$ | $\omega_{\max} + \frac{L}{\mu} + \frac{\widetilde{\mathcal{L}}_{\max}}{n\mu}$ | $\begin{cases} \omega_{\max} + \sqrt{\omega_{\max}\frac{\widetilde{\mathcal{L}}_{\max}}{n\mu}} & \text{if } nL \leq \widetilde{\mathcal{L}}_{\max} \\ \omega_{\max} + \sqrt{\frac{L}{\mu}} + \sqrt{\omega_{\max}\sqrt{\frac{\widetilde{\mathcal{L}}_{\max}}{n\mu}}\sqrt{\frac{L}{\mu}}} & \text{if } nL > \widetilde{\mathcal{L}}_{\max} \end{cases}$ |
| Iteration Complexity $\tau = d/n$ | $\frac{L_{\max}}{n\mu} + \frac{L_{\max}}{d\mu}$ (if $\nu$, $\nu_1$ are $\mathcal{O}(1)$) | $n + \frac{L_{\max}}{n\mu} + \frac{L_{\max}}{d\mu}$ (if $\nu$, $\nu_1$ are $\mathcal{O}(1)$) | $\begin{cases} n + n\left(\frac{L_{\max}}{n\mu}\right)^{1/4} & \text{if } nL \leq \widetilde{\mathcal{L}}_{\max} \\ n + \sqrt{\frac{L_{\max}}{n\mu}} + \left(n\frac{L_{\max}}{\mu}\right)^{3/8} & \text{if } nL > \widetilde{\mathcal{L}}_{\max} \end{cases}$ (if $\nu, \nu_2$ are $\mathcal{O}(1)$ and $L_{\max}/\mu$ is $\mathcal{O}(nd^2)$) |
| Reference | Theorem 2, Remark 3 | Theorem 3, Remark 4 | Theorem 4, Remark 5 |
| Speedup factor (up to) | $\min(n, d)$ | $\min(n, d)$ | $\begin{cases} \sqrt{d} & \text{if } nL \leq \widetilde{\mathcal{L}}_{\max} \text{ and } L_{\max}/\mu = \mathcal{O}(nd^2) \\ \sqrt{\min(n,d)} & \text{if } nL > \widetilde{\mathcal{L}}_{\max} \text{ and } L_{\max}/\mu = \mathcal{O}(nd^2) \end{cases}$ |

In this paper, we argue that when training supervised models, *smoothness matrices* (see Definition 1)—information-rich generalizations of the classical and ubiquitous smoothness constants—can and should be exploited for further dramatic gains, both in theory and practice.

**3.1. Unbiased diagonal sketches.** We study unbiased diagonal sketches, defined as follows:

**Definition 2** (Unbiased diagonal sketch)**.** Let $S$ be a random subset of the set of coordinates/features of the model $x \in \mathbb{R}^d$ we wish to train, i.e., $S \subseteq [d] := \{1, 2, \ldots, d\}$. Let $S$ be *proper*, i.e., $p_j := \mathrm{Prob}(j \in S) > 0$ for all coordinates $j \in [d]$. We now define a random diagonal matrix (sketch) $\mathbf{C} = \mathbf{C}_S \in \mathbb{R}^{d \times d}$ via

$$\mathbf{C} = \mathrm{Diag}(\mathrm{c}_1, \ldots, \mathrm{c}_d), \quad \mathrm{c}_j = \begin{cases} 1/p_j & \text{if } j \in S, \\ 0 & \text{otherwise.} \end{cases} \tag{6}$$

Note that given a vector $x = (x_1, \ldots, x_d) \in \mathbb{R}^d$, we have $(\mathbf{C}x)_j = \begin{cases} x_j/p_j & \text{if } j \in S \\ 0 & \text{if } j \notin S \end{cases}$. So, we can control the sparsity level of the product $\mathbf{C}x$ by engineering the properties of the random set $S$. Also note that $\mathbb{E}[\mathbf{C}x] = x$ for all $x$.

**3.2. Data-dependent sparsification operators.** In order to further alleviate the communication burden inherent in distributed optimization, we further propose *data-dependent sparsification operators* that can take full advantage of the smoothness matrices $\mathbf{L}_i$ associated with the local losses $f_i$. To the best of our knowledge, this is in sharp contrast with the design of all existing tractable compression techniques used in distributed training, which are proposed independently of the training data, and typically based on intuitive or information-theoretic principles.

With each node $i$ we associate an unbiased diagonal matrix $\mathbf{C}_i$ of the form (6). We use this and the smoothness matrix of $f_i$ to define a sparsification technique, described next.

**Definition 3** (Data-dependent sparsification). In situations when the $i$-th node wished to communicate local gradient $\nabla f_i(x)$, we ask the node to send the sparse (=compressed) vector $\mathbf{C}_i\mathbf{L}_i^{\dagger 1/2}\nabla f_i(x)$ to the server instead. The server then constructs (=decompresses) an unbiased estimator of $\nabla f_i(x)$ as:

$$g_i(x) = \mathbf{L}_i^{1/2}\mathbf{C}_i\mathbf{L}_i^{\dagger 1/2}\nabla f_i(x), \tag{7}$$

where $\mathbf{L}_i^{\dagger 1/2}$ denotes the square root of the Moore-Penrose pseudoinverse of $\mathbf{L}_i$.

Notable differences of our proposed communication protocol when compared with standard sparsification techniques are: i) we use the smoothness matrix $\mathbf{L}_i$, ii) the compressed vector $\mathbf{C}_i\mathbf{L}_i^{\dagger 1/2}\nabla f_i(x)$ is not unbiased, iii) we devise a separate decompression mechanism (7), also involving $\mathbf{L}_i$, and this enforces effective unbiasedness.

**3.3. Matrix-smoothness-aware redesign of 3 methods.** To showcase the power of our approach, we demonstrate how our matrix-smoothness-aware sparsification technique (7) can be adapted to DCGD, DIANA and ADIANA, in each case leading to significant communication savings. By doing so, we show that matrix smoothness can be effectively used to speed up communication compression, variance reduction and acceleration, respectively. This results in three novel methods: DCGD+, DIANA+, and ADIANA+; see Table 1.

**3.4. Dramatic improvements in complexity results.** We perform complexity analyses for our methods and derive convergence rates under matrix smoothness[4] (see Assumption 3) and strong convexity assumptions (see Theorems 2, 3 and 4). We show that new methods always outperform the originals/baselines, and often dramatically so.

To illustrate the potential of our sparsification technique (7) embedded in the new methods, let all machines $i \in [n]$ use sketches $\mathbf{C}_i$ induced by independent[5] samplings $S_i$ with probabilities $p_{i;j} := \mathrm{Prob}(j \in S_i)$. Then we show that, with optimized probabilities $p_{i;j}$, DCGD+ can be $\mathcal{O}(\min(n, d))$ times faster then DCGD (see Remark 3) and DIANA+ can be $\mathcal{O}(\min(n, d))$ times faster than DIANA (see Remark 4), depending on matrices $\mathbf{L}_i$. For the accelerated method, we highlight improvements when condition numbers of subproblems are $\mathcal{O}(nd^2)$. We show that ADIANA+ can be faster than the original ADIANA by a factor of $\mathcal{O}(\sqrt{d})$ in high compression regime, and by a factor of $\mathcal{O}(\sqrt{\min(n, d)})$ in low compression regime (see Remark 5). Main theoretical results are summarized in Table 2.

**3.5. Single node case.** Specializing our theory to the single machine setting ($n = 1$), we design new non-distributed algorithms providing an alternative viewpoint to randomized coordinate descent methods (see Appendix E).

**3.6. Lower bounds.** Using matrices as linear compression operators, we further investigate the trade-off between communicated bits and variance induced by the compression (see Appendix F).

**3.7. Experiments.** We conduct numerical experiments using LibSVM datasets [Chang and Lin, 2011], confirming the effectiveness and superiority of our sparsification protocol (7) over the standard sparsification scheme (see Section 6 and Section C).

# 4 New Communication-Efficient Methods Exploiting Matrix Smoothness

Consider the problem (1) with the smoothness Assumption 1 and for strongly convex $f$.

**Assumption 2** ($\mu$-convexity). $f \colon \mathbb{R}^d \to \mathbb{R}$ is $\mu$-convex for some $\mu > 0$, i.e., for all $x, y \in \mathbb{R}^d$

$$f(x) \geq f(y) + \langle \nabla f(y), x - y \rangle + \tfrac{\mu}{2}\|x - y\|^2.$$

We present our new distributed methods, redesigned for matrix smoothness, and their convergence guarantees. Each node $i \in [n]$ generates diagonal sketches $\mathbf{C}_i$ independently from others via an

---

[4]The closest to our result is work of Hanzely and Richtárik [2019b] and their ISEGA method which is able to exploit *diagonal* smoothness matrices. To the best of our knowledge, we are the first to fully exploit smoothness matrices of arbitrary structure, and elevate them as a new tool at the disposal of algorithm designers.

[5]Sampling $S_i$ is called independent if $p_{i;jl} := \mathrm{Prob}(\{j, l\} \subseteq S_i) = p_{i;j}p_{i;l}$ for all $j, l \in [d]$ such that $j \neq l$.

arbitrary sampling $S_i$ and, togther with its smoothness matrix $\mathbf{L}_i$, composes the compression matrix $\mathbf{C}_i\mathbf{L}_i^{\dagger 1/2}$. Probability matrices $\mathbf{P}_i$, $\widetilde{\mathbf{P}}_i$ associated with the sampling $S_i$ and sketch $\mathbf{C}_i$ are defined as

$$\mathbf{P}_i = (p_{i;jl})_{j,l=1}^d, \ \widetilde{\mathbf{P}}_i = (\widetilde{p}_{i;jl})_{j,l=1}^d \qquad p_{i;jl} = \mathrm{Prob}(\{j,l\} \subseteq S_i), \ \widetilde{p}_{i;jl} = \frac{p_{i;jl}}{p_{i;jj}p_{i;ll}} - 1. \quad (8)$$

Next, we introduce the key quantity, $\widetilde{\mathcal{L}}_{\max}$, describing the joint contribution of our sparsification (7) to the complexities of the three proposed methods:

$$\widetilde{\mathcal{L}}_{\max} = \max_{1 \le i \le n} \widetilde{\mathcal{L}}_i, \qquad \widetilde{\mathcal{L}}_i = \lambda_{\max}(\widetilde{\mathbf{P}}_i \circ \mathbf{L}_i), \quad (9)$$

Above, $\circ$ stands for the Hadamard (i.e. element-wise) product.

## 4.1 DCGD+

We now present our matrix-smoothness-aware sparsification technique by adapting DCGD algorithm [Khirirat et al., 2018]. Upon receiving the current model $x^k$ from the server, each node computes $\mathbf{L}_i^{\dagger 1/2}\nabla f_i(x^k)$ based on local training data and smoothness matrix. Next, sparsified updates $\mathbf{C}_i^k\mathbf{L}_i^{\dagger 1/2}\nabla f_i(x^k)$ are sent back to the server, which then averages decompressed updates $\mathbf{L}_i^{1/2}\mathbf{C}_i^k\mathbf{L}_i^{\dagger 1/2}\nabla f_i(x^k)$ and performs proximal step to get a new model $x^{k+1}$.

---
**Algorithm 1** DCGD+
---
1: **Input:** Initial point $x^0 \in \mathbb{R}^d$, current point $x^k$, step size $\gamma$, diagonal sketch $\mathbf{C}_i^k$
2: **on** server
3:     send $x^k$ to all nodes
4:     get sparse updates $\mathbf{C}_i^k\mathbf{L}_i^{\dagger 1/2}\nabla f_i(x^k)$ from each node
5:     $x^{k+1} = \mathrm{prox}_{\gamma R}(x^k - \gamma g^k)$, where $g^k = \frac{1}{n}\sum_{i=1}^n \mathbf{L}_i^{1/2}\mathbf{C}_i^k\mathbf{L}_i^{\dagger 1/2}\nabla f_i(x^k)$
---

With this method we get convergence up to a neighborhood due to compressoin noise.

**Theorem 2** (see G.3). *Let Assumptions 1 and 2 hold and assume that each node generates its own diagonal sketch $\mathbf{C}_i$ independently from others. Define $\sigma^* := \frac{1}{n}\sum_{i=1}^n \widetilde{\mathcal{L}}_i\|\nabla f_i(x^*)\|_{\mathbf{L}_i^\dagger}^2$. Then, for the step-size $0 < \gamma \le \frac{1}{L + \frac{2}{n}\widetilde{\mathcal{L}}_{\max}}$, the iterates $\{x^k\}$ of Algorithm 1 satisfy*

$$\mathbb{E}\left[\|x^k - x^*\|^2\right] \le (1 - \gamma\mu)^k \|x^0 - x^*\|^2 + \frac{2\gamma\sigma^*}{\mu n}. \quad (10)$$

## 4.2 Variance reduction: DIANA+

Next, we apply our sparsification technique to the variance reduced method DIANA [Mishchenko et al., 2019]. In this method, each node maintains an auxiliary control vector $h_i^k$, called shift, which helps to reduce the variance coming from the sparsification. Moreover, the central server keeps track of only the averaged shift $h^k$. Then, the model $x^k$ as well as control vectors $h_i^k$, $h^k$ are updated by decompressing sparse information $\Delta_i^k$ using matrices $\mathbf{L}_i$.

---
**Algorithm 2** DIANA+
---
1: **Input:** Initial point $x^0 \in \mathbb{R}^d$, initial shifts $h_i^0 \in \mathrm{Range}(\mathbf{L}_i)$, current point $x^k$, step size parameter
    $\gamma$ and $\alpha$, sketch $\mathbf{C}_i^k$ and $\overline{\mathbf{C}}_i^k := \mathbf{L}_i^{1/2}\mathbf{C}_i^k\mathbf{L}_i^{\dagger 1/2}$, current shifts $h_1^k, \ldots, h_n^k$ and $h^k := \frac{1}{n}\sum_{i=1}^n h_i^k$.
2: **on** each node
3:     get $x^k$ from the server
4:     send sparse update $\Delta_i^k = \mathbf{C}_i^k\mathbf{L}_i^{\dagger 1/2}(\nabla f_i(x^k) - h_i^k)$
5:     update local gradient and shift $\overline{\Delta}_i^k = \mathbf{L}_i^{1/2}\Delta_i^k$, $g_i^k = h_i^k + \overline{\Delta}_i^k, h_i^{k+1} = h_i^k + \alpha\overline{\Delta}_i^k$
6: **on** server
7:     get sparse updates $\Delta_i^k$ from each node
8:     $\overline{\Delta}^k = \frac{1}{n}\sum_{i=1}^n \overline{\Delta}_i^k = \frac{1}{n}\sum_{i=1}^n \mathbf{L}_i^{1/2}\Delta_i^k$, $g^k = \overline{\Delta}^k + h^k = \frac{1}{n}\sum_{i=1}^n \overline{\mathbf{C}}_i^k\left(\nabla f_i(x^k) - h_i^k\right) + h^k$
9:     $x^{k+1} = \mathrm{prox}_{\gamma R}(x^k - \gamma g^k), \quad h^{k+1} = h^k + \alpha\overline{\Delta}^k$
---

In this case we get rid of the neighborhood and provide linear convergence to the exact solution $x^*$. Throughout the paper, we use $\widetilde{\mathcal{O}}$ notation to ignore $\log \frac{1}{\varepsilon}$ factors and constants.

**Theorem 3** (see G.4). *Let Assumptions 1 and 2 hold and assume that each node generates its own diagonal sketch $\mathbf{C}_i$ independently from others. Then, for the step-size $\gamma = \frac{1}{L + \frac{6}{n}\widetilde{\mathcal{L}}_{\max}}$, Algorithm 2 guarantees $\mathbb{E}\left[\|x^k - x^*\|^2\right] \leq \varepsilon$ after*

$$\widetilde{\mathcal{O}}\left(\omega_{\max} + \frac{L}{\mu} + \frac{\widetilde{\mathcal{L}}_{\max}}{n\mu}\right) \tag{11}$$

*iterations, where $\omega_{\max} = \max_{1 \leq i \leq n} \omega_i$ and $\omega_i = \max_{1 \leq j \leq d} p_{i;j}^{-1} - 1$ is the variance of compression operator induced by sketch $\mathbf{C}_i$.*

**Remark 1** (Variance Reduction: ISEGA+). *In Appendix I we apply our redesign to another variance reduced method called ISEGA [Mishchenko et al., 2020, Hanzely and Richtárik, 2019b]. At the core of ISEGA, the mechanism for variance reduction is based on SEGA method [Hanzely et al., 2018]. The key difference between ISEGA and DIANA is that ISEGA updates the control variates $h$ more aggressively using projection instead of the mere $\alpha$-step towards the projection used in DIANA. The method is presented as Algorithm 7 in Appendix I. Theorem 22 provides the result – we can see that the worst case complexity is identical to DIANA+. However, in terms of the practical performance, we expect ISEGA+ to outperform DIANA+ due to the more aggressive update rule of control variates.*

**Remark 2** (Variance Reduction with Bi-directional Compression: DIANA++). *As an extension to DIANA+, in Appendix J we apply our sparsification technique both for nodes and for the central server, thus compressing gradients in both directions of communication. We develop and analyze DIANA++ (see Algorithm 8), for which the central server applies compression in its turn with sketch $\mathbf{C}$ independently. To converge in a linear rate, DIANA++ maintains an additional control vector to reduce the variance coming from the master's sparsification. Our convergence theory (Theorem 23) recovers the same complexity (11) of DIANA+ if no compression is applied by the master.*

### 4.3 Acceleration with variance reduction: ADIANA+

Finally, we redesign the accelerated method ADIANA [Li et al., 2020] to effectively exploit local smoothness matrices. The algorithm develops four sequences $\{x^k, y^k, z^k, w^k\}$ of models, which are layered via convex combinations, proximal steps and probabilistic assignments. In each iteration, nodes receive models $x^k$ and $w^k$ from the server, and send back sparse updates $\Delta_i^k$ and $\delta_i^k$ using local data and control vectors $h_i^k$. Then, decompressing these sparse vectors with matrices $\mathbf{L}_i$, nodes update their shifts $h_i^k$ and the server updates all four models along with averaged shift $h^k$.

---

**Algorithm 3** ADIANA+

---

1: **Input:** Initial points $x^0 = y^0 = z^0 = w^0 \in \mathbb{R}^d$, initial shifts $h_i^0 \in \text{Range}(\mathbf{L}_i)$, current point $x^k$, parameters $\gamma, \alpha, \beta, \eta, \theta_1, \theta_2, q$, sketch $\mathbf{C}_i^k$ and $\overline{\mathbf{C}}_i^k := \mathbf{L}_i^{1/2}\mathbf{C}_i^k\mathbf{L}_i^{\dagger 1/2}$, current shifts $h_1^k, \ldots, h_n^k$ and $h^k = \frac{1}{n}\sum_{i=1}^n h_i^k$
2: **on** server
3:      $x^k = \theta_1 z^k + \theta_2 w^k + (1 - \theta_1 - \theta_2)y^k$, send $x^k$ and $w^k$ to all nodes
4: **on** each node
5:      send sparse updates $\Delta_i^k = \mathbf{C}_i^k\mathbf{L}_i^{\dagger 1/2}(\nabla f_i(x^k) - h_i^k)$ and $\delta_i^k = \mathbf{C}_i^k\mathbf{L}_i^{\dagger 1/2}(\nabla f_i(w^k) - h_i^k)$
6:      update local gradient $\overline{\Delta}_i^k = \mathbf{L}_i^{1/2}\Delta_i^k$, $g_i^k = h_i^k + \overline{\Delta}_i^k$ and shift $\overline{\delta}_i^k = \mathbf{L}_i^{1/2}\delta_i^k$, $h_i^{k+1} = h_i^k + \alpha\overline{\delta}_i^k$
7: **on** server
8:      get sparse updates $\Delta_i^k$ and $\delta_i^k$ from each node
9:      $\overline{\Delta}^k = \frac{1}{n}\sum_{i=1}^n \mathbf{L}_i^{1/2}\Delta_i^k, \quad \overline{\delta}^k = \frac{1}{n}\sum_{i=1}^n \mathbf{L}_i^{1/2}\delta_i^k, \quad g^k = \overline{\Delta}^k + h^k$
10:      $h^{k+1} = h^k + \alpha\overline{\delta}^k, \quad y^{k+1} = \text{prox}_{\eta R}(x^k - \eta g^k)$
11:      $z^{k+1} = \beta z^k + (1 - \beta)x^k + \frac{\gamma}{\eta}(y^{k+1} - x^k), \quad w^{k+1} = \begin{cases} y^k & \text{with probability } q, \\ w^k & \text{with probability } 1-q. \end{cases}$

---

Clearly, ADIANA+ enjoys the accelerated rate, which is strictly better then the one for DIANA+.

**Theorem 4** (see G.5). *Let Assumptions 1 and 2 hold and assume that each node generates its own diagonal sketch $\mathbf{C}_i$ independently from others. Then, the iteration complexity of Algorithm 3*

*guaranteeing* $\mathbb{E}\left[\|z^k - x^*\|^2\right] \leq \varepsilon$ *is*

$$
\begin{cases}
\widetilde{\mathcal{O}}\left(\omega_{\max} + \sqrt{\omega_{\max}\dfrac{\widetilde{\mathcal{L}}_{\max}}{\mu n}}\right) & \text{if} \quad nL \leq \widetilde{\mathcal{L}}_{\max} \\[3ex]
\widetilde{\mathcal{O}}\left(\omega_{\max} + \sqrt{\dfrac{L}{\mu}} + \sqrt{\omega_{\max}\sqrt{\dfrac{\widetilde{\mathcal{L}}_{\max}}{\mu n}}\sqrt{\dfrac{L}{\mu}}}\right) & \text{if} \quad nL > \widetilde{\mathcal{L}}_{\max}.
\end{cases}
\tag{12}
$$

# 5 Improvements Over the Original Methods

To compare the proposed methods with originals and highlight improvement factors, we choose independent sampling for all nodes. For Algorithms 1 and 2, we optimize probabilities of the samplings based on the complexities we found.

**5.1. Parameters describing matrices** $\mathbf{L}_i$**.** Define parameters $\nu$, $\nu_s$ describing local smoothness as

$$
\nu := \frac{\sum_{i=1}^n L_i}{\max_{i \in [n]} L_i}, \quad \nu_s := \max_{i \in [n]} \frac{\sum_{j=1}^d \mathbf{L}_{i;j}^{1/s}}{\max_{j \in [d]} \mathbf{L}_{i;j}^{1/s}},
\tag{13}
$$

where $L_i = \lambda_{\max}(\mathbf{L}_i)$ and $s = 1$ or $s = 2$. Let $L_{\max} := \max_{1 \leq i \leq n} L_i$ and $\mathbf{L}_{i;j}$ be the $j$th diagonal element of matrix $\mathbf{L}_i$. Note that parameters $\nu \in [1, n]$ and $\nu_s \in [1, d]$ describe the distribution over the nodes and coordinates respectively. If $\mathbf{L}_i$ are distributed uniformly, then $\nu = n$ and $\nu_s = d$. On the other extreme, when the distribution is extremely non-uniform, we have $\nu \ll n$ and $\nu_s \ll d$. These parameters are used to highlight the range of iteration complexities new methods can provide.

**5.2. Importance sampling for DCGD+.** Let $\tau = \mathbb{E}\left[|S_i|\right] = \sum_{j=1}^d p_{i;j}$ be the expected mini-batch size for the samplings $S_i$, where $p_{i;j} = p_{i;jj}$. Notice that convergence rate of Algorithm 1 depends on $\widetilde{\mathcal{L}}_{\max} = \max_{1 \leq i \leq n} \widetilde{\mathcal{L}}_i$. Since each node $i \in [n]$ generates its own diagonal sketch $\mathbf{C}_i$ independently from others, each node can optimize $\widetilde{\mathcal{L}}_i = \lambda_{\max}(\widetilde{\mathbf{P}}_i \circ \mathbf{L}_i)$ independently based on local smoothness matrix $\mathbf{L}_i$. In general, minimizing $\lambda_{\max}(\widetilde{\mathbf{P}}_i \circ \mathbf{L}_i)$ with respect to probability matrix $\widetilde{\mathbf{P}}_i$ is hard. However, when each node uses an independent sampling, which means $p_{i;jl} = p_{i;j}p_{i;l}$ if $j \neq l$, then

$$
\lambda_{\max}(\widetilde{\mathbf{P}}_i \circ \mathbf{L}_i) = \max_{1 \leq j \leq d}\left(\frac{1}{p_{i;j}} - 1\right)\mathbf{L}_{i;j},
\tag{14}
$$

for which we can find the optimal probabilities $p_{i;j}$. To minimize the maximum term in (14), we should have $\left(1/p_{i;j} - 1\right)\mathbf{L}_{i;j} = \rho_i$ for some $\rho_i \geq 0$. Then the solution is

$$
p_{i;j} = \frac{\mathbf{L}_{i;j}}{\mathbf{L}_{i;j} + \rho_i},
\tag{15}
$$

where $\rho_i \geq 0$ is the unique solution to $\sum_{j=1}^d \frac{\mathbf{L}_{i;j}}{\mathbf{L}_{i;j} + \rho_i} = \tau$. While the latter does not allow closed form solution for $\rho_i$, it can be computed numerically using one dimensional solvers. Thus, we can efficiently compute the optimal probabilities (15).

**Proposition 5** (Optimality). *The independent sampling with probabilities (15) is the optimal independent sampling for the rate (10).*

**Remark 3** (Improvement over DCGD [Khirirat et al., 2018]). *With probabilities (15) we show in Appendix H.1 that*

$$
\frac{L}{\mu} + \frac{\widetilde{\mathcal{L}}_{\max}}{n\mu} \leq \left(\frac{\nu}{n} + \frac{\nu_1}{\tau n}\right)\frac{L_{\max}}{\mu}.
\tag{16}
$$

*In the interpolation regime (i.e.* $\nabla f_i(x^*) = 0$ *for all* $i \in [n]$*), the iteration complexity of DCGD is* $\widetilde{\mathcal{O}}(\frac{L}{\mu} + \frac{\omega L_{\max}}{n\mu})$ *for general compression operator with variance parameter* $\omega$*. In case of sparsification, it is known that* $\tau = d/n$ *is the optimal choice under standard (scalar) smoothness assumption [Mishchenko et al., 2019]. If we specialize compression to sparsification with* $\tau = d/n$ *entries (which gives* $\omega = d/\tau - 1 = n - 1$*), we get* $\widetilde{\mathcal{O}}(\frac{L_{\max}}{\mu})$*. Notice that, in this regime, Theorem 2 also provides linear convergence with iteration complexity* $\widetilde{\mathcal{O}}(\frac{L}{\mu} + \frac{\widetilde{\mathcal{L}}_{\max}}{n\mu})$*. Based on (16), it is bounded by* $\widetilde{\mathcal{O}}((\frac{\nu}{n} + \frac{\nu_1}{d})\frac{L_{\max}}{\mu})$*, which is always better than* $\widetilde{\mathcal{O}}(\frac{L_{\max}}{\mu})$ *and can be as small as* $\widetilde{\mathcal{O}}(\frac{L_{\max}}{\min(n,d)\mu})$*. Hence, for mini-batch* $\tau = d/n$*, DCGD+ (Algorithm 1) guarantees the same* $\widetilde{\mathcal{O}}(\frac{L_{\max}}{\mu})$ *complexity in the worst case, but could provide up to* $\min(n, d)$ *times speedup. An anologous observation can be made between standard sparsificaiton with uniform probabilites and our sparsification with uniform probabilities (see Remark 7 for the details).*

**5.3. Importance sampling for DIANA+.** To find optimal probabilities for DIANA+, we minimize $\omega_{\max} + \frac{\widetilde{\mathcal{L}}_{\max}}{\mu n}$ part of the complexity (11), which is in turn equivalent to minimize

$$\max_{1 \leq j \leq d} \left( \frac{1}{p_{i;j}} - 1 \right) \mathbf{L}'_{i;j}, \quad \mathbf{L}'_{i;j} := \frac{\mathbf{L}_{i;j}}{\mu n} + 1, \tag{17}$$

which can be solved in the same way as (14) yielding

$$p_{i;j} = \frac{\mathbf{L}'_{i;j}}{\mathbf{L}'_{i;j} + \rho'_i} = \frac{\mathbf{L}_{i;j} + \mu n}{\mathbf{L}_{i;j} + (1 + \rho'_i)\mu n}. \tag{18}$$

**Proposition 6** (Optimality). *The independent sampling with probabilities (18) is the optimal[6] independent sampling for the complexity (11).*

**Remark 4** (Improvement over DIANA [Mishchenko et al., 2019, Horváth et al., 2019b]). *The iteration complexity of DIANA, with $\tau = d/n$ is $\widetilde{\mathcal{O}}(n + \frac{L_{\max}}{\mu})$. With probabilities (18) we upper bound the iteration complexity (11) in Appendix H.2 as follows*

$$\omega_{\max} + \frac{L}{\mu} + \frac{\widetilde{\mathcal{L}}_{\max}}{\mu n} \leq \frac{2d}{\tau} + \left( \frac{\nu}{n} + \frac{2\nu_1}{\tau n} \right) \frac{L_{\max}}{\mu}. \tag{19}$$

*Therefore, with $\tau = d/n$, DIANA+ (Algorithm 2) guarantees the same $\widetilde{\mathcal{O}}(n + \frac{L_{\max}}{\mu})$ complexity in the worst case, but could provide up to $\min(n, d)$ times speedup with complexity $\widetilde{\mathcal{O}}(n + \frac{L_{\max}}{\min(n,d)\mu})$.*

**5.4. Independent sampling for ADIANA+.** Clearly, if we sparsify with uniform probabilities $p_{i;j} = \tau/d$, then Algorithm 3 recovers the rate of ADIANA.

**Remark 5** (Improvement over ADIANA [Li et al., 2020]). *To show that the rate could be significantly better in some cases, consider the following choice $p_{i;j} = \sqrt{\frac{\mathbf{L}'_{i;j}}{\mathbf{L}'_{i;j} + \rho''_i}}$, $\mathbf{L}'_{i;j} = \frac{\mathbf{L}_{i;j}}{\mu n} + 1$, where $\rho''_i$ is determined uniquely from $\sum_{j=1}^{d} p_{i;j} = \tau$. Then, with these probabilities and for $L_{\max}/\mu = \mathcal{O}(nd^2)$, we show in Appendix H.3 that $\frac{L}{\mu} \leq \frac{\nu L_{\max}}{n\mu}$, $\omega_{\max} = \mathcal{O}\left( \frac{\nu_2 d}{\tau} \right)$, $\frac{\mathcal{L}_{\max}}{\mu n} = \mathcal{O}\left( \frac{\nu_2 d}{\tau} \sqrt{\frac{L_{\max}}{n\mu}} \right)$. Furthermore, assuming both $\nu$ and $\nu_2$ are $\mathcal{O}(1)$, choosing $\tau = d/n$ we get $\frac{L}{\mu} \leq \mathcal{O}\left( \frac{L_{\max}}{n\mu} \right)$, $\omega_{\max} = \mathcal{O}(n)$, $\frac{\mathcal{L}_{\max}}{\mu n} = \mathcal{O}\left( \sqrt{\frac{n L_{\max}}{\mu}} \right)$. Then, the complexity (12) of ADIANA+ reduces to*

$$\begin{cases} n + n \left( \frac{L_{\max}}{n\mu} \right)^{1/4} & \text{if } nL \leq \widetilde{\mathcal{L}}_{\max}, \\ n + \sqrt{\frac{L_{\max}}{n\mu}} + \left( n \frac{L_{\max}}{\mu} \right)^{3/8} & \text{if } nL > \widetilde{\mathcal{L}}_{\max}, \end{cases}$$

*which, compared to the complexity of ADIANA with $\omega = \mathcal{O}(n)$ compression, gives $\sqrt{d}$ times improvement in the first case and $\sqrt{\min(n, d)}$ times improvement in the second case (ignoring the first summand $n$ of the complexities).*

# 6 Experiments

We numerically compare the proposed matrix-smoothness-aware sparsification strategy (7) with the usual sparsification scheme. We devise three different experiments on logistic regression with LibSVM data [Chang and Lin, 2011]. Due to space limitations, only two of them is presented in the main part. Further numerical results alongside with their experimental details are in Appendix C.

**Experiment: Proposed and usual sparsification techniques for the 3 distributed methods.** We compare six different methods: well-established DCGD, DIANA, ADIANA and our methods DCGD+, DIANA+, ADIANA+, all with uniform sampling for $\tau = 1$. In order to highlight the importance of the variance reduction, in this experiment we choose the starting point to be close to the optimum.

Figure 1 demonstrates that: i) methods with matrix-aware sparsification (i.e., DCGD+, DIANA+, ADIANA+) outperform their baselines (i.e., DCGD, DIANA, ADIANA) ii) acceleration almost

---

[6]In the sense that it minimizes a quantity, which is the complexity of DIANA+ up to some constant factor.

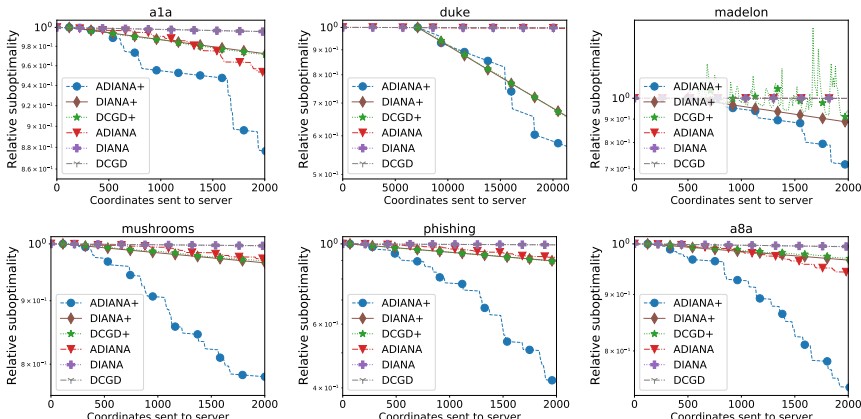

**Figure 1:** Comparison of the original methods DCGD [Khirirat et al., 2018], DIANA [Mishchenko et al., 2019] and ADIANA [Li et al., 2020] with the proposed new methods DCGD+ (Alg. 1), DIANA+ (Alg. 2) and ADIANA+ (Alg. 3). All methods use uniform sampling with $\tau = 1$.

always outperforms the non-accelerated variant, often dramatically so and iii) variance reduction never hurts the convergence, but often stabilizes the oscillation of the non-variance reduced counterpart.

**Experiment: Variance reduction with new sparsification and importance sampling.** We now comment on the experiment illustrated in Figure 2. We examine three sparsification schemes (two variants of our strategy and the usual sparsification) and their influence on convergence using six different datasets. Considered schemes are i) DIANA+ with importance sampling (18), ii) DIANA+ with uniform sampling, and iii) DIANA with uniform sampling, i.e., uniform sparsification unaware of smoothness matrices. In all three cases we fixed the sampling size $\tau = 1$.

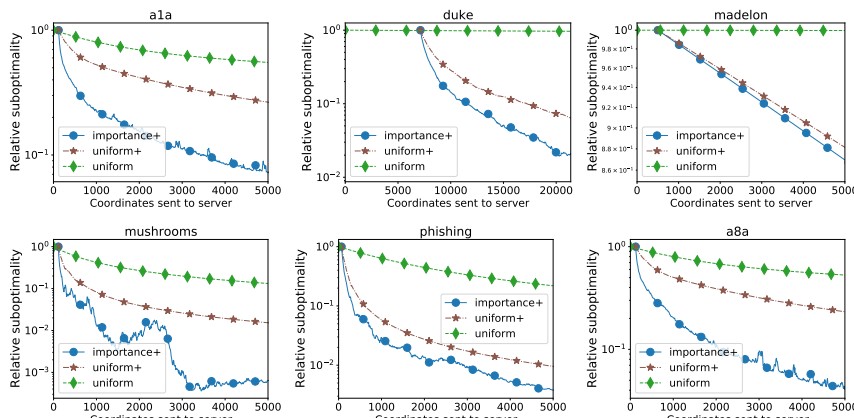

**Figure 2:** Comparison of our sparsification strategy of size $\tau = 1$ for DIANA+ (Algorithm 2) using i) importance sampling with probabilities (18), ii) uniform sampling with $p_{i;j} = \frac{1}{d}$ and iii) DIANA [Mishchenko et al., 2019] using standard sparsification scheme with uniform sampling. All methods are run with stepsizes as dictated by theory.

As expected, Figure 2 confirms our theoretical findings. First, it demonstrates that our sparsification (7) always outperforms the naive/direct sparsification, sometimes by a large margin. Second, it shows the benefit of importance sampling (18) over the uniform sampling.

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
