# OpenReview forum: "Smoothness Matrices Beat Smoothness Constants: Better  Communication Compression Techniques for Distributed Optimization"
_NeurIPS.cc/2021/Conference — NeurIPS 2021 Poster_

### Official Review · Reviewer_1hXg · 2021-07-11

**Rating:** 7
**Confidence:** 4

**Summary:**

The papers consider the standard distributed optimization scenarios, where a server wants to minimize the average of $n$ loss functions, which are evenly distributed across $n$ participating worker nodes. Typically, optimization problems designed for such distributed optimization scenarios assume a blanket smoothness constant for the average and the individual functions involved. Instead, the authors consider the notion of matrix smoothness, where the distance between the points in defining the smoothness is measured with matrix norm (of a smoothness matrix). Under the assumption that the individual loss function satisfies matrix smoothness with different matrices, the authors design compression algorithms using these smoothness matrices and then use them in conjunction with various distributed optimization algorithms. The authors show that as a result of their smoothness-aware compression, they improve over the convergence guarantees of the standard baseline algorithms considered, namely, DCGD, DIANA, ADIANA.

**Limitations And Societal Impact:**

Yes.

**Main Review:**

Originality:

  a) The notion of matrix smoothness considered in this paper is a novel one. None of the present distributed algorithms have exploited loss function smoothness to this granularity to the best of my knowledge. As the authors show, this notion can further improve the performance of distributed optimization algorithms.

  b) The compression operator proposed is tailored to the smoothness matrix of the loss function and therefore improves over the performance of generic compression operators proposed in the literature.

 c) The distributed optimization algorithms used by the authors seem to only differ in the compression part with respect to the ones present in the literature.

Presentation:

a) The paper's presentation is excellent. The introduction clearly mentions the paper's main ideas, contributions, and comparison to prior work.

 b) A minor point would be to clarify if the authors introduce the notion of matrix smoothness or was it already present in the literature. (Either way, my review would remain unchanged based on this point, but it will be instructive for the reader.)


Significance:

 a) The idea of using smoothness to this granularity is a novel one and can significantly improve the performance of distributed optimization algorithms.  However, it is unclear to me how the precise smoothness matrix could be identified for complicated loss
functions.

b) This work could also inspire other research works to explore other properties of functions to this granularity (e.g., Strong Convexity).

To summarize, I like this paper and recommend that it be accepted.



**Time Spent Reviewing:**

4

---

> ### Author Response · Authors · 2021-08-09
> **Response to Issue 1: is matrix smoothness a novel notion?**
>
> > A minor point would be to clarify if the authors introduce the notion of matrix smoothness or was it already present in the literature. (Either way, my review would remain unchanged based on this point, but it will be instructive for the reader.)
>
> Response:
>
> - The notion of *matrix smoothness* can be traced back at least to [Richtárik and Takác, 2016a], [Hanzely and Richtárik, 2019a] (mentioned in Section 2.2, line 95 in our paper) and [Qu and Richtárik 2016b] (mentioned on line 107 in our paper), where it was used in the context of *randomized coordinate descent methods*. These works were not concerned with distributed optimization, nor with communication efficiency. Instead, smoothness matrices were therein used to derive better theoretical stepsizes for parallel (minibatch) versions of randomized coordinate decent methods in an attempt to study parallelization/minibatch speedup of these methods. We do not claim to have invented this notion. We will be most happy to add these and more similar historical comments to the paper.
>
> - However, to the best of our knowledge, our work is the first that suggests that matrix smoothness can be exploited in the context of communication-efficient distributed optimization via the design of more powerful compression operators. Needless to say, exploiting matrix smoothness in the context of the sparsification was highly non-trivial. We have been thinking about this for a number of years before we found the correct solution.

---

> ### Author Response · Authors · 2021-08-09
> **Response to Issue 2: how to identify matrix smoothness for complicated losses?**
>
> > However, it is unclear to me how the precise smoothness matrix could be identified for complicated loss functions.
>
> Response:
>
> - What types of complicated losses do you have in mind?
>
> - Please recall that Lemma 1 in our paper gives a formula for computing the smoothness matrix of a function of the form $f_{i}(x)=\frac{1}{m_{i}} \sum_{m=1}^{m_{i}} \phi_{i m}\left(\mathbf{A}_{i m} x\right). $ This structure is quite universal in machine learning, as we explain in the paper.
>
> -  We did not investigate the question of computing or estimating the smoothness matrices of functions which do not arise this way. However, we believe that in such cases perhaps these matrices can be estimated heuristically throughout the algorithm run, in a similar way in which quasi-Newton methods estimate Hessian matrices from first order information on the fly. However, since we did not really put enough effort into asking these types of questions, we will have to leave extensions of his type to future research.

---

> > ### Comment · Reviewer_1hXg · 2021-08-31
> > **Thanks**
> >
> > I thank the authors for responding to my review. After reading the response and other reviews, I am happy with my previous assessment.
> >
> > Yes, I meant loss functions that cannot be described by the structural assumption in Lemma 1.

---

### Official Review · Reviewer_WW3C · 2021-07-11

**Rating:** 6
**Confidence:** 3

**Summary:**

This paper analyzes several optimization concepts: compression, variance reduction, and acceleration, using the assumption of smoothness matrices rather than the smoothness parameter. Sketching matrices are designed as sum_{i\in S} L^{1/2}_i c_i * L^{1/2,dagger}_i where L_i is the "smoothness matrix" associated with the ith data sample. Taking this additional information into account, which for generalized linear models is not too hard to compute, it is suggested that there can be a factor of min(n,d) improvement in unaccelerated methods and sqrt(min(m,n)) benefit in accelerated methods.

**Ethical Concerns:**

No ethical concerns

**Limitations And Societal Impact:**

No negative societal impact

**Main Review:**

This work is interesting, and though the contribution feels incremental, the L_i matrices are indeed not complicated to precompute for most applications. Overall, the writing is fairly clean and the results do seem novel, though it is unclear to me if they are a bit incremental, as it seems to be not that much of an improvement over using leverage scores.

The one big weakness is that it seems most of the paper is actually in a 50 page appendix, which is not just extra experiments and a few lengthy proofs, but quite a bit of clarifying text and interpretations as well. Moreover many of the proofs pushed to the appendix are short, so it is not the lengthiness of the proofs that is causing such a big appendix. The impact is that while I can check a few pages of the appendix, I cannot reasonably judge all 50 pages in this short review period, nor do I think that is fair compared to the other works that squeezed all the big ideas in the first 8 pages. At the very least, the authors should condense their main story to clearly and succinctly outline how to get their main constant factor improvements. That being said, the parts I was able to read seem reasonable; the proof techniques seem standard and the results follow expected trends.

Specific critiques:
 - The definition of L_i in (5) should be rank m_i for problems like logistic regression, but could be of higher rank in a neural network model. These two regimes seem a bit separate: sketching should make a big impact in shallow models (like logistic regression) but in the extreme case of m_i = 1, L_i matrix shouldn't be that much more handy than L_i scalar, and it seems that this is somewhat the example shown in the experimental results. On the other hand, a deep model that can overfit may have a more complex L_i. Am I misinterpreting this? Are the experiments using L_i that have more interesting rank? These details should be included in the main paper, not in the appendix.

 - Are the results significantly better than using leverage scores? What are the main scenarios where this method would give benefit?

 - Can the authors point to a specific scenario where L_i the matrix is giving a significant benefit over L_i a smoothness constant? e.g. give a specific sparsification scheme that is used in practice, or a dataset type with such features in the data matrix?

**Time Spent Reviewing:**

2

---

> ### Author Response · Authors · 2021-08-09
> **Issue 1: 50 page appendix**
>
> > The one big weakness is that it seems most of the paper is actually in a 50 page appendix, which is not just extra experiments and a few lengthy proofs, but quite a bit of clarifying text and interpretations as well. Moreover many of the proofs pushed to the appendix are short, so it is not the lengthiness of the proofs that is causing such a big appendix. The impact is that while I can check a few pages of the appendix, I cannot reasonably judge all 50 pages in this short review period, nor do I think that is fair compared to the other works that squeezed all the big ideas in the first 8 pages. At the very least, the authors should condense their main story to clearly and succinctly outline how to get their main constant factor improvements. That being said, the parts I was able to read seem reasonable; the proof techniques seem standard and the results follow expected trends.
>
> Response:
>
> We *strongly disagree* with the claim that most of the paper is in the  50p appendix and that we did not manage to squeeze all of our key ideas within the page limit. We believe that our results presented within the main body are complete on their own. While most of the extra material provided in the appendix is actually *not necessary*, we still believe it provides *extra value* for readers who want to understand more, and for the experts in the field. We believe we could easily cut at least 25 pages of the appendix without hurting the paper quality significantly. However, we do not want to do so as these pages do bring some *extra value* for certain readers.
>
> Let us go through the individual appendix sections to explain why we do not think that the reviewer's criticism is justified:
> - Section A contains a conclusions and extensions text. Many excellent papers do not contain such a section at all; or only include a brief one. Again, our paper does not stand or fall on the inclusion of this section. We believe it could be useful to some readers, however.
> - Section B lists the limitations of our work.  We are happy to move this whole section to the main body of the paper if the reviewer wishes so (we sketched some limitations therein already).
> - Section C contains extra experiments. It is common practice for NeurIPS papers to include some extra experiments in the appendix. You should feel free to ignore these in your evaluation of our paper. However, we should not be penalized for including extra experiments. We are not writing our paper for the reviewers, but for the readers who wish to use or build upon our work. This extra material can be very useful to them.
> - Section D is a notation table. We provide it as a *courtesy* to the reader. This is a good thing, and of course is not required nor necessary.  Should we be penalized for including it? Of course not.
> - The whole Section E is completely unnecessary for the story of our paper. Indeed, in it we develop some of our main results in the simple single node scenario. We do so because this material could be easier to digest for readers less familiar with the topics of our paper. So, we provide value to certain types of readers, and we exerted effort to do so.
> - Section F contains nothing but a formal algorithm statement of our method ADIANA+. The technical pseudocode itself is not important at first reading  as we already stated and described the non-accelerated  DCGD+ and DIANA+ in the main body. So, we felt it was OK to omit the pseudocode of ADIANA+ from the main body of the paper.
> - Section G discusses the general theoretical limits of linear compressors. These results are orthogonal to the main results of our paper, and thus could be easily omitted. However, we believe that they can serve as a good starting point for researchers who want to push the limits of sparsification methods further.
> - Section H contains the proofs of the theorems. It is customary for NeurIPS papers to include proofs in the appendix.
> - Section I contains derivation of the claims from Section 5. While we could state the whole of Section 5 as a theorem, and replace Section I with the proof of the theorem, we believe that our current approach is cleaner.
> - Section J introduces an alternative approach to variance reduction (for dealing with variance introduced by sparsification) that is different to Diana. Again, this extension merely points to what can be done further with our main results. Our paper dos not stand or fall on this. Again, you can feel free to ignore this additional tangential contribution if you wish so. Again, however, we should not be penalized for presenting an additional contribution in the appendix, one that is not central to the main/key developments of our paper.
> - Section K presents an extension of our main methods, one that demonstrates *bi-directional* sparsification. You can feel free to ignore this additional contribution. However, we should not be penalized for having an additional contribution.
>
>
> As we hope you can see, all content we included in the appendix belongs to one of these categories:
> - Content that is traditionally/ typically relegated to the appendix in NeurIPS papers. This includes extra experiments (Sec C), details that are not needed for the clear flow of ideas in the main body of the paper (Sec F), and formal proofs (Sec H, Sec I).
> - Courtesy to the reader - notation table (Sec D). This is normally not included at all, but if it is, it certainly belongs to the appendix.
> - Further thoughts that sometimes are but often are not included in NeurIPS papers, such as conclusions and extensions (Sec A) and limitations (Sec B). We would be quite happy to move much of this text to the main body of the paper if he had a bit extra space (for instance, in the camera-ready version of the paper).
> - Extensions (Sec K), tangential developments (Sec G, Sec J), and pedagogical simplifications (Sec E) that could in principle be completely removed from the paper since they do not support any of the main claims. Also, these developments can be ignored by the reviewers. However, we *did* obtain these *extra* results in the course of our research, and we *do* believe that they may be of use to some readers. So, we find it better to include them than not.
>
> The reviewer also said that
>
> >  Moreover many of the proofs pushed to the appendix are short, so it is not the lengthiness of the proofs that is causing such a big appendix.
>
> Indeed, some of the proofs are short, and some of them only seem to be short as they are compactly partitioned into several lemmas with short proofs to facilitate the reading process. Again, this should certainly not count as a weakness. We hope that our explanation above of what is contained within each section in the appendix settles your concerns.
>
> We will now offer a few further comments related to this portion of the concern:
> > The impact is that while I can check a few pages of the appendix, I cannot reasonably judge all 50 pages in this short review period, nor do I think that is fair compared to the other works that squeezed all the big ideas in the first 8 pages. At the very least, the authors should condense their main story to clearly and succinctly outline how to get their main constant factor improvements.
>
> Please note that we *did* manage to include all our key ideas and results in the main body of the paper. These are:
> - (i) when training supervised models in the distributed setup, smoothness matrices can and should be exploited for further dramatic gains when available,
> - (ii) there is a novel communication sparsification strategy that can take full advantage of the smoothness matrices associated with local losses, and
> - (iii) our new sparsification technique can be adapted to three distributed optimization algorithms.
>
> These are the big ideas of our work. All are properly motivated, included and presented in the first 9 pages. The other two reviewers seem to be fine with the writing of the paper. For example, reviewer 1hXg wrote
>
> >> The paper's presentation is excellent. The introduction clearly mentions the paper's main ideas, contributions, and comparison to prior work.
>
> and reviewer zeTh wrote
>
> >> The paper is reasonably well-written.
>
> We as authors are also very happy with the writing. The content in the many appendices is either standard (e.g., proofs), or entirely optional and tangential, and not critical.

---

> > ### Comment · Reviewer_WW3C · 2021-08-19
> > **response**
> >
> > To clarify, I'm not bringing this up to say that only proofs can be 50 pages, or that proofs that are 50 pages are fun to read. I'm just pointing out that usually when an appendix is very long, it's to accommodate something like a proof that is too long and burdensome to put in the main paper, and doesn't really add interest to most readers. To me, as a reviewer, it seemed like a good chunk of this appendix was meant to add to the quality of the paper, and I'm surprised the authors chose to not showcase it, but I am not judging that beyond saying that I did not read it carefully and don't feel its contributions should be a huge part of the decision. If that is ok with the authors, then it's completely fine.

---

> > > ### Author Response · Authors · 2021-08-19
> > > **Thanks for the clarification**
> > >
> > > We are on the same page then.
> > >
> > > There is some potentially very useful material in the appendix, as we explained. But we believe the main contributions *are* explained in the main paper. **Feel free *not* to read the appendix carefully and judge our paper based on the main contributions only.**
> > >
> > > But please also keep in mind that we designed our appendix to be helpful to the interested reader who wants to work on these or similar ideas - the appendix does add value to the paper in the sense we explained above (e.g., notation table is helpful, simplified proof for $n=1$ case can be helpful to someone not used to distributed methods, detailed proofs of the main theorems are there, extra experiments, ...).

---

> ### Author Response · Authors · 2021-08-09
> **Issue 2: Structure and rank of the smoothness matrices**
>
> > The definition of $L_i$ in (5) should be rank $m_i$ for problems like logistic regression, but could be of higher rank in a neural network model. These two regimes seem a bit separate: sketching should make a big impact in shallow models (like logistic regression) but in the extreme case of $m_i = 1$, $L_i$ matrix shouldn't be that much more handy than $L_i$ scalar, and it seems that this is somewhat the example shown in the experimental results. On the other hand, a deep model that can overfit may have a more complex $L_i$. Am I misinterpreting this? Are the experiments using $L_i$ that have more interesting rank? These details should be included in the main paper, not in the appendix.
>
> Response:
>
> - This criticism is partially related to the first limitation of our approach discussed in our Section B. In case you missed this, please can you have a look there?
>
> - We do not expect our method to be practical when the number $d$ of model parameters is enormously large, and when the smoothness matrices $\boldsymbol{L}_i$ are dense, of high rank, and  without any spacial structure. We admit that practicality of our approach, at least at the current level of the development of our ideas (perhaps many of these limitations can be lifted y us or others in the future - this is the beauty of research),  is limited to either the case when $d$ is not too large, or when the matrices  $\boldsymbol{L}_i$ are of special structure (e.g. low rank, diagonal).
>
> - Why do you think that the extreme case of $m_i = 1$, rank-1 smoothness matrices may not be beneficial ? Please note that the theory is the same, and Figure 4 confirms this numerically.

---

> > ### Comment · Reviewer_WW3C · 2021-08-19
> > **response**
> >
> >  - Why do you think that the extreme case of $m_i = 1$, rank-1 smoothness matrices may not be beneficial ? Please note that the theory is the same, and Figure 4 confirms this numerically.
> >
> > Well, the question is how well it compares against just knowing the element-wise Lipschitz constant, as is practice in, say,  coordinate descent methods.

---

> > > ### Author Response · Authors · 2021-08-19
> > > **Comparing diagonal and rank-1 smoothness matrices**
> > >
> > > First, notice that both diagonal and rank-1 smoothness matrices are just a special case of this more general treatment. These two cases have similar properties but none of them can be claimed to always outperform the other based on our theory. In more details, the key factors that we care in our work related to smoothness matrices are the followings:
> > >
> > > - ***How efficient local smoothness matrices can be obtained ?***
> > >
> > > ***This depends on the structure of the model***. For instance, we discussed generalized linear models for which smoothness matrices can be computed directly and explicitly using the training data.
> > >
> > > - ***How much storage and communication cost is required to store and send those matrices to the server ?***
> > >
> > > Both diagonal and rank-1 matrices require ***the same*** $O(d)$ storage and communication cost for the clients. The server also needs $O(nd)$ storage to maintain all matrices for both cases.
> > >
> > > - ***How many flops are required for the server to perform a single iteration ?***
> > >
> > > Recall that the server needs to multiply $\mathbf L_i^{1/2}$ by sparse updates $\mathbf C_i^k \mathbf L_i^{\dagger 1/2}\nabla f_i(x^k)$ in each iteration for all devices $i\in[n]$. For diagonal smoothness matrices $\mathbf L_i$ all these operations can be done in a straightforward manner in $O(d)$ evaluation cost for each device. Furthermore, the same evaluation cost is enough for rank-1 smoothness matrices since both square root and pseudo-inverse of a rank-1 matrix admit closed form solutions. In particular, if $u$ is a column-vector, then
> > > $(u u^\top)^{1/2} =  \frac{u u^\top}{\\|u\\|}$, and
> > > $(u u^\top)^{\dagger} = \frac{u u^\top}{\\|u\\|^4}$.
> > > With this observation, note that for rank-1 matrices, the server needs the same $O(d)$ flops to perform the multiplication for each device.
> > > Thus, both diagonal and rank-1 matrices require ***the same*** amount of flops to perform a single iteration.
> > >
> > >
> > > - ***What are the values of parameters $\nu$ and $\nu_s$ describing smoothness matrices ?***
> > >
> > > These are the *key metrics* of smoothness matrices governing theoretical properties of the proposed methods. Smaller the values of $\nu$ and $\nu_s$ are, better theoretical guarantees can be established. To understand which structure (diagonal or rank-1) gives better bounds, we need to estimate those parameters $\nu$ and $\nu_s$ for those cases. Thus, whether diagonal matrices are more beneficial than rank-1 smoothness matrices ***depends on the entries of those smoothness matrices***. For example, all values $\nu\in[1,n]$ and $\nu_s\in[1,d]$ are feasible for diagonal/rank-1 smoothness matrices.

---

> ### Author Response · Authors · 2021-08-09
> **Issue 3: Leverage scores**
>
> > Overall, the writing is fairly clean and the results do seem novel, though it is unclear to me if they are a bit incremental, as it seems to be not that much of an improvement over using leverage scores.... Are the results significantly better than using leverage scores? What are the main scenarios where this method would give benefit?
>
> Response:
>
>
> - We are not aware of any alternative approach to do what we are doing in our paper: we are not aware of any prior work that proposes the use of smoothness matrices as objects of key interest in communication-efficient distributed training, and proposes their use in the design of smoothness-matrix-aware bespoke compression operators.
>
> - You do not make it clear what approach you have in mind when mentioning *leverage scores*. Unfortunately, we can't answer your question as the question is not clear. Please can you explain what you mean? Can you cite the relevant work you want us to compare to?
>
> - We argue that it is not reasonable to be asked to respond to criticism that is not properly explained. We are happy to respond once the suggestion is made clear. Otherwise, we respectfully request that this criticism be dropped.

---

> > ### Comment · Reviewer_WW3C · 2021-08-19
> > **response**
> >
> >  - We are not aware of any alternative approach to do what we are doing in our paper: we are not aware of any prior work that proposes the use of smoothness matrices as objects of key interest in communication-efficient distributed training, and proposes their use in the design of smoothness-matrix-aware bespoke compression operators.
> >
> > Yes, but the idea of using quadratic approximations is a fundamental tool in optimization. To me, this seems like a global Hessian approximation, which is a fairly commonly used trick. Anyway I'm not saying I've seen this exact implementation, but at the very least the work could be presented in the context of other Hessian approximation tools, such as Newton subsampling, or Kernel subsampling.
> >
> >  - You do not make it clear what approach you have in mind when mentioning leverage scores.
> >
> > Leverage scores are commonly used to approximate kernel matrices, for example in row/column sampling for Nystrom sampling. There are many papers on this. (e.g. Gittens, Mahoney 2013)

---

> > > ### Author Response · Authors · 2021-08-19
> > > **2 points**
> > >
> > > Thanks for your response; we are glad that you are engaging with us as this gives us the chance to clarify these points.
> > >
> > > Your first point:
> > >
> > > - Yes, the idea of approximation with (convex) quadratics that have arbitrary (pos semidefinite) Hessians, rather than just multiples of identity, is a classical idea many areas of optimization, including sequential quadratic programming and quasi-Newton methods. **However, to the best of our knowledge, it was *not* used in the context of communication-efficient distributed optimization, and it was *not* used to design more potent compressors.** We thus believe that this is not valid criticism. Indeed, this is a generic criticism that can be applied to virtually any paper. For example, one could criticize interior point methods for relying on Newton's steps (which are known), Nesterov's acceleration for merely applying the idea of (Polyak) momentum in a different way, every paper on gradient methods for using the idea of fixed point iterations and so on. What we are trying to say is: it is important to look at the differences as well, not just at the similarities.
> > >
> > > - Indeed, please note that **there is a *very long distance* between the *idea* of applying matrix smoothness to communication efficient distributed optimization, and discovering the *correct way* to do it.** It is not at all clear how this should be done. We have thought about this problem on and off for 2-3 years before we realized what the solution should look like, and even the at that point was a long way to performing the theoretical analysis.
> > >
> > > - Our work is only very loosely related to Hessian approximation. For instance, our techniques apply to functions that have Lipschitz gradient, but are not necessarily twice differentiable. We do not perform any explicit Hessian approximation. For functions with a bounded Hessian (in some weighted norm), one can indeed think of smoothness matrices as providing a uniform bound on the Hessian. But such an approximation would be very loose since, for example, it does not change with the iterates. So, we do not believe thinking of smoothness matrices as Hessian approximation is particularly insightful. **Nevertheless, we will add a paragraph ruling this connection.**
> > >
> > >
> > > Your second point:
> > >
> > > - We know what leverage scores *are*, and are familiar with the second paper you cite. However, we do not understand the link you are trying to point to between our work and leverage scores since you did not explain it. You are merely suggesting there is a link, or perhaps even suggesting leverage scores can be used to do what we are doing (design gradient compressors for distributed training) - but you did not provide any explanation, nor a link to any paper that would explain the link. **So, this criticism is not intelligible to us. Please can you explain it *in detail*? We would be happy to look into this and respond. However, we suggest that if an explanation is not provided, this criticism should be dropped as it is not substantiated.**
> > >
> > > We are looking forward to more discussion. We hope to get the chance to clarify all your points! Please do not hesitate to ask anything.

---

> ### Author Response · Authors · 2021-08-09
> **Issue 4: Specific scenario when smoothness matrices beat smoothness constants**
>
> > Can the authors point to a specific scenario where ${\bf L}_i$ the matrix is giving a significant benefit over $L_i$ a smoothness constant? e.g. give a specific sparsification scheme that is used in practice, or a dataset type with such features in the data matrix?
>
> Response:
>
> - The simplest example would be when the matrices ${\bf L}_i$'s are all *diagonal*.  Such a scenario might either occur naturally, or alternatively we can reduce any situation to it by upper bounding the smoothness matrix via a diagonal matrix, which will generally be tighter than upper bounding it by a single constant (times the identity). In such a case, implementing our sparsification strategy requires almost no communication/computation overhead. At the same time, our theory still suggests significant improvement over naive sparsification in terms of the iteration complexity, at most of order $\min(n, d)$, depending on the data.
>
> - Of course,  there are many more scenarios where our sparsification would beat the baselines.

---

> > ### Comment · Reviewer_WW3C · 2021-08-19
> > **response**
> >
> > A diagonal sparsification sounds a lot like the block-specific $L_i$ constants often used in block coordinate descent methods. (See Wright 2015 survey.) I agree it is useful, but it is not first presented in this work.

---

> > > ### Author Response · Authors · 2021-08-19
> > > **Yes and No**
> > >
> > > Yes, there is a close connection - but we already made this clear in our paper, and also in our response to a different reviewer entitled "Response to Issue 1: is matrix smoothness a novel notion?". Please read this response, copy-pasted here for convenience:
> > >
> > > > The notion of matrix smoothness can be traced back at least to [Richtárik and Takác, 2016a], [Hanzely and Richtárik, 2019a] > (mentioned in Section 2.2, line 95 in our paper) and [Qu and Richtárik 2016b] (mentioned on line 107 in our paper), where it was used in the context of randomized coordinate descent methods. These works were not concerned with distributed optimization, nor with communication efficiency. Instead, smoothness matrices were therein used to derive better theoretical stepsizes for parallel (minibatch) versions of randomized coordinate decent methods in an attempt to study parallelization/minibatch speedup of these methods. We do not claim to have invented this notion. We will be most happy to add these and more similar historical comments to the paper.
> > >
> > > > However, to the best of our knowledge, our work is the first that suggests that matrix smoothness can be exploited in the context of communication-efficient distributed optimization via the design of more powerful compression operators. Needless to say, exploiting matrix smoothness in the context of the sparsification was highly non-trivial. We have been thinking about this for a number of years before we found the correct solution.
> > >
> > > Diagonal matrix smoothness matrices are just a special case of this more general treatment. We already admit that the more general treatment was used in the literature of coordinate descent methods.
> > >
> > > We believe that the transfer of an idea from one subfield to another where it was not used before is a valid and potent approach to research. We did not simply just apply existing results from coordinate descent literature to our setting. We merely used the notion of matrix smoothness used there; all else is new.

---

### Official Review · Reviewer_zeTh · 2021-08-03

**Rating:** 6
**Confidence:** 3

**Summary:**

This paper proposes a new notion of smoothness called matrix smoothness which leads to better theory bounds and a data-dependent compression scheme. The compression scheme is applied to three existing algorithms resulting in improved convergence bounds in smooth and strongly convex setting. Simple experiments are done to show the effectiveness of the proposed scheme.

**Limitations And Societal Impact:**

I have concerns about the feasibility of the proposed method. First, I expect it’d be hard to estimate the matrices $\mathbb{L}_i$ especially in high-dimensional deep learning settings. Second, modern ReLU networks don’t satisfy smoothness so this method may not be applicable there. It’d be nice to know if it can be extended for non-convex and non-smooth deep learning problems. The proposed compression scheme would probably increase per-client or per-device computation because you need to invert the $L_i$’s.

**Main Review:**

The idea of using smoothness matrices to improve convergence bounds as well as perform data-dependent compression seems novel but kind of infeasible for large-scale problems (see next section). Theoretically, the proposed scheme seems to offer a significant speed-up when importance sampling is used. The paper is reasonably well-written. Some comments:
1) Since you consider strongly convex and smooth objectives, it’d be good to show how much speed-up you get in practice vs. how much you expect theoretically.
2) Is it possible to give an estimate of the speed-up when you use uniform sampling and not importance sampling?
3) In Theorem 2 for DCGD+, sigma^{*} can become large if $L_i^{*}$ is poorly conditioned (then the max eigenvalue of its inverse will become large)?
4) line 195 - typo “compression”
Please see my concerns in the next section.

**Time Spent Reviewing:**

2

---

> ### Author Response · Authors · 2021-08-09
> **Response to Issue 1: speedup in practice vs theory**
>
> > Since you consider strongly convex and smooth objectives, it'd be good to show how much speed-up you get in practice vs. how much you expect theoretically.
>
> Response:
>
> - We *have* been comparing the theoretical vs the experimental speedup when we conduced our experiments as a part of sanity check. The empirical speedup was indeed matching the empirical one after a certain number of iterations. This is no surprise as we used the theory supported stepsizes in each case, and these stepsize are key to determine the empirical convergence.
>
> - We initially decided to omit the comparison of theoretical vs the practical speedup as we already have a lot of content in the paper. However, we are most happy to include such plots in the camera-ready version of the paper.

---

> ### Author Response · Authors · 2021-08-09
> **Response to Issue 3: increased neighborhood in Theorem 2**
>
> > In Theorem 2 for DCGD+, $\sigma^*$ can become large if ${\bf L}_i^\dagger$ is poorly conditioned (then the max eigenvalue of its inverse will become large)?
>
> Response:
>
> This is a valid observation but not that important as we already have a great way to get around it in the paper! Indeed, the convergence neighborhood might get bigger when our sparsification strategy is used, and one can think of this as paying a cost of a larger neighborhood for obtaining a faster initial convergence rate.  However, this is not an issue as we show, via the DIANA+ method, that this neighborhood can be  removed  completely, preserving the same (fast) linear convergence speed.

---

> ### Author Response · Authors · 2021-08-09
> **Response to Issue 4: typo on line 195**
>
> > line 195 - typo “compression” Please see my concerns in the next section.
>
> Response:
>
> Typo is fixed. Thank you!

---

> ### Author Response · Authors · 2021-08-09
> **Response to Issue 5: feasibility of the proposed method**
>
> > I have concerns about the feasibility of the proposed method. First, I expect it'd be hard to estimate the matrices especially in high-dimensional deep learning settings. Second, modern ReLU networks don't satisfy smoothness so this method may not be applicable there.
>
> Response:
>
> - There *is* a range of applications where smoothness matrices are not hard to estimate (Lemma 1 shows one such generic example for generalized linear models).
>
> - Note that we do not claim that the proposed method would be practical for high-dimensional deep learning problems (see the first point in section B). This is not our object of study, and hence this criticism is not justified. If feasibility means applicability to non-convex and non-smooth problems in high-dimensional deep learning, then most of the optimization papers submitted and accepted to conferences like NeurIPS, ICML, ICLR are infeasible (almost all papers assume some form of smoothness condition, for example). Of course, it is important to understand deep learning applications. But making this as an entry requirement is not justified. Besides, how can we expect to understand complex problems such as deep learning without having a deep understanding of simpler problems? Understanding the simpler problems forms the solid basis of probing more complicated applications. This is how science works.

---

> ### Author Response · Authors · 2021-08-09
> **Response to Issue 6: extension to deep learning**
>
> > It'd be nice to know if it can be extended for non-convex and non-smooth deep learning problems.
>
> Response:
>
> As this is the *first* time that the notion of matrix smoothness exploited in distributed optimization, there are obviously many possible extensions one an try to work on, including
> - non-convex problems,
> - subsampling of the local objective and stochastic gradients,
> - extending to other compression schemes and
> - other optimization methods,
> - reducing local computation
>
> In the appendix we investigated a few directions to show the applicability of our approach. However, it is not reasonable to study all these cases in one paper. Science works step by step.
>
> We do believe our work will lead to many extensions, but we can't study them all in one paper. It takes a community to push this agenda!
>
> While extending our results to the non-convex problems could be viable, the non-smooth relaxation seems a bit questionable as the idea of our work is to exploit and utilize smoothness information of the problem. Therefore,  we do not expect a non-smooth theory that would lead to as strong conclusions as we have viable.
>
> Nonetheless,  we believe that our results might serve as a starting point to the development of an efficient heuristic for  deep learning. For example, one might try to learn the smoothness structure of the objective along the optimization path and then plug it into our methods, resulting in possibly very efficient heuristics.  There are many other approaches that one can explore that would be tailored specifically for deep learning; however, this is beyond the scope of our paper.

---

> ### Author Response · Authors · 2021-08-09
> **Response to Issue 7: more computation since matrices need to be inverted**
>
> > The proposed compression scheme would probably increase per-client or per-device computation because you need to invert the ${\boldsymbol L}_i$'s.
>
> Response:
>
> - Indeed, we discuss the possible computation overhead in section B, devoted to limitations. Note that those matrix inversions are done *only once*, before the training process.
>
> - In general, there is a trade-off between compression and computation. However, the selection of ${\boldsymbol L}_i$'s is purely up to the user. So, one can choose how much of the the extra computation they are willing to tolerate.  As we describe in the paper, there are many examples where one gets plenty of communication savings for almost no extra computation paid.

---

> ### Author Response · Authors · 2021-08-09
> **Response to Issue 2: speedup for uniform sampling**
>
> > Is it possible to give an estimate of the speed-up when you use uniform sampling and not importance sampling ?
>
> Response:
>
> - This is a great question! In fact, it is possible to give similar estimate in that case too.
>
>
> For standard uniform sparsification, the term affected by the compression in the complexity (consider the linear rate of DCGD for simplicity) is $\omega L_{\max} = (\frac{d}{\tau}-1)L_{\max}$, where $L_{\max} = \max_{i\in[n]} L_i$ is the largest smoothness constant over devices. On the other hand, in the proposed sparsification strategy we have probabilities $p_{i;jl} = \frac{\tau^2}{d^2}$ if $j\ne l$, and $p_{i;jl} = \frac{\tau}{d}$ if $j=l$, which implies that $\widetilde{\mathbf P}_i = ( \frac{d}{\tau}-1 ) \textbf{I}$. In this case, the term affected by the compression in the complexity is
>
> $\widetilde{\cal L}_{\max}$
>
> $= \max_{i\in[n]} \lambda_{\max} (\widetilde{\\mathbf P}_i \circ \\mathbf L_i) $
>
> $= \(\frac{d}{\tau}-1\) \max_{i\in[n]}\lambda_{\max}( \textbf{Diag}(\\mathbf L_i)) $
>
> $= \(\frac{d}{\tau}-1\) \mathbf L_{\max}$,
>
>
> where $\mathbf L_{\max} = \max_{i\in[n]}\max_{j\in[d]} \mathbf L_{i;j}$ is the largest diagonal element over all smoothness matrices.
> Now notice that $\mathbf L_{\max} \le L_{\max} \le d \mathbf L_{\max}$
> hold and bounds are tight. Hence, the upper bound obtained for our sparsification is always better and can be up to $d$ times better depending on the ratio $\frac{L_{\max}}{\textbf{L}_{\max}} \in [1,d]$.
>
> Thus, we can make an analogous observation between classical uniform sampling and our uniform sampling albeit with a different condition on smoothness matrices, i.e. $\frac{L_{\max}}{\textbf{L}_{\max}} = \Omega(d)$ instead of $\nu_s = {\cal O}(1)$.

---

### Author Response · Authors · 2021-08-09
**Thanks to all reviewers!**

Dear reviewers,

Thanks for reading our manuscript and for offering your comments and suggestions. We are grateful for your time and effort.

We wish to summarize the positive comments you have collectively made in your reports:

- "The idea of using smoothness matrices to improve convergence bounds as well as perform data-dependent compression seems novel" [Reviewer zeTh]
- "Theoretically, the proposed scheme seems to offer a significant speed-up when importance sampling is used." [Reviewer zeTh]
- "The paper is reasonably well-written." [Reviewer zeTh]
- "This work is interesting" [Reviewer WW3C]
- "Overall, the writing is fairly clean and the results do seem novel" [Reviewer WW3C]
- "the parts I was able to read seem reasonable" [Reviewer WW3C]
- "The notion of matrix smoothness considered in this paper is a novel one. None of the present distributed algorithms have exploited loss function smoothness to this granularity to the best of my knowledge. As the authors show, this notion can further improve the performance of distributed optimization algorithms." [Reviewer 1hXg]
- "The compression operator proposed is tailored to the smoothness matrix of the loss function and therefore improves over the performance of generic compression operators proposed in the literature." [Reviewer 1hXg]
- "The paper's presentation is excellent. The introduction clearly mentions the paper's main ideas, contributions, and comparison to prior work."  [Reviewer 1hXg]
- "The idea of using smoothness to this granularity is a novel one and can significantly improve the performance of distributed optimization algorithms."  [Reviewer 1hXg]
- "This work could also inspire other research works to explore other properties of functions to this granularity (e.g., Strong Convexity)"  [Reviewer 1hXg]
- "To summarize, I like this paper and recommend that it be accepted." [Reviewer 1hXg]

You have of course also raised a few critical comments, asked a few questions, and made a few suggestions. We have done our best to address them. As we hope you will see from our replies, all issues raised were minor, or not issues. We therefore believe we successfully addressed all your concerns, and if you also think so, we would kindly wish to ask you to consider raising your scores appropriately.

Should any issues remain, please do let us know. We will be most happy to reply.

---

### Decision · Program_Chairs · 2021-09-27

**Decision:**

Accept (Poster)

**Comment:**

The authors are in general positive agreement as to both the theoretical and empirical contributions of this paper. I would recommend this be accepted.